# Statins—Beyond Their Use in Hypercholesterolemia: Focus on the Pediatric Population

**DOI:** 10.3390/children11010117

**Published:** 2024-01-17

**Authors:** Elena Lia Spoiala, Eliza Cinteza, Radu Vatasescu, Mihaela Victoria Vlaiculescu, Stefana Maria Moisa

**Affiliations:** 1Department of Pediatrics, Faculty of Medicine, “Gr. T. Popa” University of Medicine and Pharmacy, 700115 Iasi, Romania; elena-lia.spoiala@umfiasi.ro (E.L.S.); stefana-maria.moisa@umfiasi.ro (S.M.M.); 2Department of Pediatrics, “Carol Davila” University of Medicine and Pharmacy, 020021 Bucharest, Romania; 3Department of Pediatric Cardiology, “Marie Curie” Emergency Children’s Hospital, 041451 Bucharest, Romania; 4Cardio-Thoracic Department, “Carol Davila” University of Medicine and Pharmacy, 020021 Bucharest, Romania; 5Clinical Emergency Hospital, 014461 Bucharest, Romania; 6“Diabnutrimed” Medical Center, 020359 Bucharest, Romania; vvmihaela@yahoo.com; 7“Sfanta Maria” Clinical Emergency Hospital for Children, 700309 Iasi, Romania

**Keywords:** dyslipidemia, statins, children

## Abstract

Statins are a class of medications primarily used in adults to lower cholesterol levels and reduce the risk of cardiovascular events. However, the use of statins in children is generally limited and carefully considered despite the well-documented anti-inflammatory, anti-angiogenic, and pro-apoptotic effects, as well as their effect on cell signaling pathways. These multifaceted effects, known as pleiotropic effects, encompass enhancements in endothelial function, a significant reduction in oxidative stress, the stabilization of atherosclerotic plaques, immunomodulation, the inhibition of vascular smooth muscle proliferation, an influence on bone metabolism, anti-inflammatory properties, antithrombotic effects, and a diminished risk of dementia. In children, recent research revealed promising perspectives on the use of statins in various conditions including neurological, cardiovascular, and oncologic diseases, as well as special situations, such as transplanted children. The long-term safety and efficacy of statins in children are still subjects of ongoing research, and healthcare providers carefully assess the individual risk factors and benefits before prescribing these medications to pediatric patients. The use of statins in children is generally less common than in adults, and it requires close monitoring and supervision by healthcare professionals. Further research is needed to fully assess the pleiotropic effects of statins in the pediatric population.

## 1. Introduction

Statins primarily function by inhibiting 3-hydroxy-3-methylglutaryl-coenzyme A (HMG-CoA) reductase, a key enzyme involved in cholesterol biosynthesis [1]. However, beyond their lipid-lowering effects, statins possess pleiotropic properties that may be relevant to various pediatric diseases [2]. These include anti-inflammatory, anti-angiogenic, and pro-apoptotic effects, as well as the modulation of cell signaling pathways [3]. These multifaceted effects, known as pleiotropic effects, encompass enhancements in endothelial function, a significant reduction in oxidative stress, the stabilization of atherosclerotic plaques, immunomodulation, the inhibition of vascular smooth muscle proliferation, an influence on bone metabolism, anti-inflammatory properties, antithrombotic effects, and a diminished risk of dementia. Beyond their established benefits, statins have been investigated for their potential applications in other life-threatening conditions such as cancer and inflammatory bowel disease [4].

This narrative review aims to highlight the various effects and indications of statins in pediatrics. We focus on updates regarding classic indications, as well as innovative potential uses in various conditions including neurological, cardiovascular, and oncologic diseases, as well as special situations such as transplanted children.

## 2. Methods

We performed an electronic literature search using the terms “statins”, “children”, “paediatric”, “dyslipidemia”, and “hypercholesterolemia”. The PubMed and Embase databases were systematically searched from the time of their inception until December 2023. The reference list of the articles was carefully reviewed as a potential source of information. Only articles in English were selected in our analysis. Overall, 88 manuscripts were screened by 2 independent authors. As in all narrative reviews, a selection bias cannot be excluded. 

## 3. The Use of Statins in Hypercholesterolemia

Severe hypercholesterolemia is characterized by high levels of total cholesterol (TC) primarily due to elevated LDL cholesterol (LDL-C) exceeding the 95th percentile or 190 mg/dL. The prolonged elevation of LDL-C is considered a risk factor for the development of atherosclerotic cardiovascular disease, particularly ischemic heart disease. Familial hypercholesterolemia (FH) is a well-studied form of severe HC, caused by a major variant in one gene (LDLR, APOB, PCSK9, or ApoE), with an autosomal codominant pattern of inheritance [5]. These genetic variants lead to extremely high LDL-C levels and the early onset of ischemic heart disease. 

However, a significant portion of severe HC cases, known as polygenic hypercholesterolemia (PH), can be attributed to the cumulative effect of multiple single nucleotide variants scattered throughout the genome [5]. These individuals are at a high risk of developing atherosclerosis due to lifelong exposure to elevated LDL-C levels [6]. Therefore, early intervention is crucial in managing this condition and reducing the risk of complications.

For children diagnosed with heterozygous familial hypercholesterolemia (FH), guidelines suggest considering the initiation of statin therapy between from the age of 8 and 10 years [7]. Before starting statins, it is important to rule out any secondary causes of dyslipidemia and ensure that LDL cholesterol (LDL-C) levels remain persistently elevated despite 3–6 months of lifestyle modifications [7]. In cases where children with heterozygous FH have more severe LDL-C abnormalities, statin treatment may be initiated alongside therapeutic lifestyle changes [8]. 

The therapy initiation algorithm is based on fasting LDL-C values and the presence or absence of cardiovascular risk factors. Children and adolescents with LDL-C ≥ 250 mg/dL and/or triglycerides (TG) ≥ 500 mg/dL should be evaluated by a specialist physician. Patients with lower LDL-C values but a significant family history of premature cardiovascular disease (stroke, acute coronary events, or peripheral artery disease in males younger than 45 and females younger than 55 years of age), as well as any child for whom medication treatment may be indicated (LDL-C ≥ 160 mg/dL with risk factors (obesity, hypertension, type 1 or 2 diabetes, metabolic syndrome) or ≥190 mg/dL without risk factors), should benefit from medication intervention [9].

The first-line drug is statin, starting at the lowest dose [10]. According to Elkins et al., the expected lipid level values in the pediatric population differ from those in adults and vary depending on age [8]. Table 1 provides a brief overview of the posology and the percentage of LDL-C reduction in children with familial hypercholesterolemia.

Statins have been shown to effectively reduce LDL cholesterol (LDL-C) levels in children and adolescents with hypercholesterolemia (Table 1). A recent meta-analysis comprising data from eight randomized controlled trials and 1025 children with familial hypercholesterolemia demonstrated that the lipid profile was significantly improved after statin therapy with a mean reduction of 75 mg/dL in the case of TC, 11.5 mg/dL in the case of TG, and 75 mg/dL in the case of LDL-C, with a mean increase of 14.3 mg/dL in the case of HDL-C [17]. In another comprehensive meta-analysis of 10 randomized controlled trials involving 1191 children aged 13.3 ± 2.5 years, statins have shown effectiveness in reducing TC by 25% and TG by 8% while increasing HDL-C by 3% when compared with a placebo [18].

While statins are generally well tolerated, there are some potential safety concerns to be aware of. Common side effects of statin use in children may include gastrointestinal symptoms such as abdominal pain, nausea, or diarrhea [19]. Muscle-related side effects, such as myalgia or elevated creatine kinase levels, can also occur but are rare in children [20]. Serious adverse events associated with statin use, such as liver dysfunction or rhabdomyolysis, have been also rarely reported in the pediatric population [20]. According to Khoury et al., liver toxicity, myositis, and rhabdomyolysis were no more frequently reported in children receiving statins than in those receiving placebos and had no impact on growth or development [21]. However, in the case of pre-existing liver disease, such as non-alcoholic fatty liver disease, liver function and transaminase levels should be monitored more closely when initiating statin therapy [22]. A fasting lipid-profile should be recommended 4 to 8 weeks after statin initiation; if adequate LDL-C reduction is observed, a fasting lipid profile should be repeated every 3–6 months in the first 112 months and longitudinally every 6–12 months [23]. Transaminases and creatine phosphokinase levels should be checked at baseline and also in the case of new symptoms [23]. Following the initiation of statin therapy, it is recommended to regularly monitor liver enzymes and creatine phosphokinase levels at 1–2 months, assess liver enzymes at 3–6 months, and periodically thereafter. It is important to note that the routine monitoring of creatine phosphokinase is not necessary as it may lead to incidental elevations of muscular enzymes that could be attributed to physical activity [23].

Up to the present, no cases of rhabdomyolysis were reported in children [24]. In a cohort of 1501 children aged 14 ± 4 years old treated with one of two low dose statins (simvastatin or atorvastatin), no muscle symptoms or clinically significant increase in creatin kinase levels were observed [20]. However, in case of myopathy, it is recommended to promptly measure creatine phosphokinase levels. If there are slight elevations in liver enzymes or creatine phosphokinase, it is not necessary to discontinue statin treatment. If transaminases increase more than 3-fold and creatine phosphokinase levels increase more than 10-fold, it is advised to promptly discontinue statins and investigate other potential causes of muscular and liver dysfunction [22].

Although there have been theoretical concerns about potential disruptions to steroid synthesis pathways in young individuals receiving statins, clinical trials have not shown any significant impact on hormone pathways, pubertal development, or growth. In a 10-year follow-up study, Kusters et al. demonstrated that the use of statins in children with familial hypercholesterolemia was not associated with modifications regarding growth velocity, sexual maturation, and Tanner staging [25].

As can be seen in Table 2, the benefits of statin therapy in children generally outweigh the potential risks. The most notable study of these follow-up analyses reveals that even after 20 years from the initiation of statin therapy, there were no reported cases of rhabdomyolysis or any other serious adverse events during the study [12]. Moreover, all the cited studies showed that statins did not influence the growth or the development of the included studies.

In the present, the use of statins in children is generally limited and carefully considered due to potential safety concerns [24]. There is limited research on the long-term safety and efficacy of statins in pediatric populations. Most studies have focused on adults, and the impact of these medications on growing children is not fully understood. Due to their possible adverse events, statins should be carefully prescribed based on a risk-benefit assessment, taking into account the child’s overall health, the severity of the underlying condition, and the potential risks associated with the medication. 

Regarding drug interactions, it is important to consider that statins can interact with other medications and that their plasma concentrations can be raised as a consequence of competing substrates, increasing both their lipid-lowering effects and risk for adverse events [22]. One important exception is the most frequent anticoagulant used in pediatrics, warfarin [28], as simvastatin, atorvastatin, and rosuvastatin were reported to potentiate its effects [27]. As lovastatin, simvastatin, and atorvastatin are substrates of cytochrome P450 3A4 (CYP3A4), among other medications, macrolides, antifungal azoles, protease inhibitors, cyclosporine, and grapefruit should be avoided, with dose adjustments often needed for calcium channel blockers and amiodarone [22]. In the case of fluvastatin, pitavastatin, and rosuvastatin, which are substrates of CYP2C9, cyclosporine should be avoided [22]. As pravastatin is the only statin not metabolized by a CYP isoenzyme, it may be a more appropriate choice for patients at risk for adverse drug interactions [22]. Also, the concurrent use of statins and gemfibrozil should be avoided, as gemfibrozil inhibits OATP1B1, a hepatic drug membrane transporter, and can hinder the hepatic glucuronidation of statins, leading to elevated plasma concentrations of statins and their metabolites [22].

While specific prevalence rates may vary across different regions and populations, studies have indicated an overall increase in the prevalence of lipid metabolism disorders in children, not only as a consequence of an increasing number of primary hypercholesterolemia cases, but also due to an epidemic of secondary hypercholesterolemia cases [29]. The various causes of secondary hypercholesterolemia are detailed in Table 3. 

In diabetes, the role of HMG-CoA reductase inhibitors is controversial and unclear [30]. Joyce et al. reported a positive association with an increased likelihood of developing type 2 diabetes in children without dyslipidemia who took statins, not observed in statin-treated dyslipidemia children [31]. Unfortunately, this article fails to explicitly identify the actual indications for statin use in the non-dyslipidemia group, and only compares the outcomes in the statin-exposed and non-exposed groups in terms of type 2 diabetes occurrence. In this respect, one needs to keep in mind the rapid progression of adolescent- and young adult-onset type 2 diabetes [32] and the high number of complications and severe metabolic phenotypes, as shown by studies such as TODAY and RISE [33,34,35].

## 4. Statins and Cardiovascular Disorders

Although cited as a secondary cause of hypercholesterolemia, the association between dyslipidemia and atherosclerosis in Kawasaki disease is still not certain [36]. In this context, we will further discuss the use of statins in Kawasaki disease due to their pleiotropic effects.

Kawasaki disease, also known as Kawasaki syndrome, is an acute febrile illness of unknown cause primarily observed in children under the age of 5. While it is generally considered a rare disease [37], it can have serious complications if left untreated due to the development of coronary artery abnormalities, which can lead to coronary artery aneurysms, depressed myocardial contractility and heart failure, myocardial infarction, and arrhythmias [38]. 

As statins have been found to have positive effects on inflammation, endothelial function, and oxidative stress, recommendations from the American Heart Association [39,40] and Japanese Circulation Society [41] encourage the empirical use of statins for children with Kawasaki disease and past or current aneurysms. 

In this context, it is important to understand the mechanisms beyond the positive impact on inflammation and coronary artery abnormalities. In a murine study, atorvastatin demonstrated anti-atherosclerotic effects by enhancing the function of endothelial cells in artificially induced Kawasaki-like vasculitis [42]. Motoji et al. showed that the use of statins may be able to prevent the cardiovascular events associated with Kawasaki disease by stimulating the expression of endothelial nitric oxide synthase (eNOS), which restored the function of endothelial cells [42]. In children, atorvastatin safety was investigated by Niedra et al. in a study on 20 patients with postacute Kawasaki disease complicated by a coronary artery aneurysm [43]. The authors observed that the use of atorvastatin for a median of 30 months led to a decrease of 14% in the case of TC, 20% in the case of LDL-C, and 5% in the case of TG [42]. Interestingly, HDL-C recorded a relative decrease of 4% [42]. Only minor adverse events (seven children with a transient mild increase of liver enzymes and one child with high creatine phosphokinase levels) and no apparent adverse effect on growth or development were reported [42]. 

The safety of atorvastatin was also investigated in a Phase I/IIa 2-center dose-escalation study including 34 children aged 2–17 years old with Kawasaki disease complicated by a coronary artery aneurysm [44]. The authors concluded that after 6 weeks of treatment with 0.75 mg/kg/day of atorvastatin, no serious adverse events were reported [44]. Additionally, a small study including 13 male children aged 2–10 years found that 6 months of pravastatin therapy improved chronic vascular inflammation and endothelial dysfunction in children with Kawasaki disease with minimal to no adverse effects [45].

Besides Kawasaki disease, the statins showed potential benefits in chronic vasculitis such as Behçet’s and rheumatoid arthritis due to their anti-inflammatory, anti-oxidant, and endothelial-repairing properties [46]. Recent research showed that the anti-inflammatory effects of statins are based on complex mechanisms: statins can suppress TLR4 (Toll-like receptor 4)/MyD88/NF-ĸB (Nuclear factor kappa-light-chain-enhancer of activated B cells) signaling and cause an immune response shift to an anti-inflammatory response and also inhibit the NF-ĸB pathway by decreasing the expression of TLRs 2 and 4 [47]. Several potential mechanisms on how statins impact TLR-signaling pathways have been proposed: the inhibition of protein prenylation, direct or indirect NF-ĸB inhibition, the inhibition of the MyD88/NF-ĸB pathway, and the enhancement of anti-inflammatory response elements [48]. As a result, the statins may lead to atherosclerotic plaque stabilization [47]. Another proposed mechanism in combating local inflammation is decreased protein prenylation as a consequence of the ability of statins to decrease oxidized LDL (oxLDL) stabilization [47]. However, further studies are needed to unravel the potential use of these mechanisms in pediatric cardiovascular and inflammatory diseases. As statins exhibit both a lipid-lowering effect in addition to an anti-inflammatory effect [49], their use was also investigated in heart-transplanted patients. In adults, statins have been found to have several beneficial effects [50], and early use reduced the cardiac allograft vasculopathy and the risk of the accelerated progression of atherosclerosis [51]. In heart-transplanted children, the pleiotropic effects of statins are still understudied. Moreover, the results of the available studies on the impact of statins in this particular population show controversial and even conflicting results. In a study comprising 78 children aged 4.8 to 14.7 years old who underwent a heart transplant, the use of statins was associated with a significant decrease in the incidence of acute cellular rejection and post-transplant lymphoproliferative disease, but with no significant decrease in the risk of coronary artery vasculopathy even with early statin initiation [52]. In a retrospective study including 964 heart transplant recipients aged 5–18 years old, Greenway et al. reported no effects of statin use regarding overall survival up to 5 years post-transplant and the risk of cardiac allograft vasculopathy and post-transplant lymphoproliferative disease [53]. These results are concordant with a recent study conducted by Townsend et al. in 3485 pediatric heart transplant recipients, showing no statistically significant difference regarding the graft survival and the incidence of cardiac allograft vasculopathy between children receiving consecutive statin therapy and those with intermediate use or no statin therapy [54]. Interestingly, pravastatin effectively lowered the TC and LDL-C and improved the compositional properties of LDL and HDL (2) particles, but failed to normalize the elevated TG and prevent the progression of transplant vasculopathy in 19 pediatric cardiac transplant recipients aged 4 to 18 years old [55]. To sum up, the use of statins in cardiac transplant recipients is not universally accepted or standardized, but the field of transplant medicine is dynamic, and ongoing research may provide more insights into the role of statins in improving the outcomes for heart-transplanted children. Due to their anti-inflammatory [56], immunomodulatory [57], antioxidant [58], and antithrombotic effects [59], as well as due to their benefits on the endothelial function [60], HMG-CoA inhibitors should be the topic of future research in order to evaluate the effects of statin therapy on post-transplant outcomes, such as graft survival, rejection rates, and overall patient morbidity and mortality.

## 5. Statins and Neurological Disorders

While the primary indication for statin therapy is the management of dyslipidemia and prevention of cardiovascular disease, there is growing interest in exploring their potential neuroprotective properties. In the last decades, statins have been investigated for their potential effects on neurodevelopmental disorders. 

Recent research indicated that statins could potentially improve outcomes in children with autism spectrum disorder (ASD) due to their anti-inflammatory properties. In a randomized, double-blind study, Moazen-Zadeh et al. reported a positive impact on irritability and hyperactivity/noncompliance in children aged 4–12 years old treated with risperidone (1–2 mg/day) associated with simvastatin (20–40 mg/day) than in those treated with risperidone alone [61]. However, no significant difference was observed in lethargy/social withdrawal, stereotypic behavior, and inappropriate speech [61]. Another randomized controlled trial conducted by Stivaros et al. [62] in children with autism associated to type 1 neurofibromatosis with a mean age of 8.1 years old revealed that simvastatin exerted specific effects in key brain areas which were highly associated with social impairment and autism psychopathology [63]. These results may be explained by the neurobiological findings in children with autism which indicate a disruption in the balance of excitatory and inhibitory neurotransmission, lipid metabolism, and immune/inflammatory responses [64]. Although very promising, the use of statins in children with autism is not specifically approved for this indication by regulatory authorities. Further research is needed to better understand the potential benefits and risks, as well as the optimal dosing regimens of statin therapy in this condition. 

The evidence regarding the use of statins in Duchenne muscular dystrophy (DMD, the most common hereditary neuromuscular disease) is limited and even conflicting. In a murine study, Mucha et al. reported that simvastatin did not improve the clinical and biological patterns of *mdx* (Male dystrophin-deficient) mice and, moreover, it was even responsible for an increased rate of muscular necrosis [65]. In contrast, Whitehead et al. suggested that statins may help mitigate the progression of muscle damage by reducing inflammation and oxidative stress, which are key contributors to muscle degeneration [66]. Moreover, there is evidence that simvastatin improved diastolic function and prevented cardiac fibrosis in *mdx* mice, which suggests that statins may be an important contributor to the treatment of DMD-associated cardiomyopathy [67]. Genetic studies also confirmed that cholesterol metabolism, especially via the mevalonate pathway, is a potential therapeutic target in DMD [68]. However, it is important to note that the use of statins in DMD is still under investigation, and their efficacy and safety in human patients with DMD have not been established.

The use of statins has been also investigated in neurological symptoms associated with genetic syndromes, such as Smith—Lemli—Opitz syndrome [69], Fragile X syndrome, and Rett syndrome. In Smith—Lemli—Opitz syndrome, a genetic disorder characterized by impaired cholesterol synthesis due to a deficiency of the enzyme 7-dehydrocholesterol reductase, which leads to reduced levels of cholesterol and an accumulation of its precursor, 7-dehydrocholesterol, the role of statins is controversial [70]. Ballout et al. systematically assessed the available evidence on the efficacy of statins on the neurobehavioral abnormalities of these patients and concluded that although statins can indirectly increase cholesterol levels by inhibiting the HMG-CoA reductase, there is no clear evidence of the positive impact on survival, quality of life, and reduction in irritability [69]. In Fragile X syndrome, known as the most common inherited cause of intellectual disability and autism in children, Çaku et al. showed a significant statistical improvement for hyperactivity, lethargy, social avoidance, and stereotypy after three months of lovastatin in escalating doses (up to 40 mg) in a study of 15 patients aged between 6 and 31 years old [71]. The role of statins in Rett syndrome, an X-linked disease characterized by the progressive development of neurological and motor dysfunction, was suggested by a murine study conducted by Buchovecky et al., in which statins had a favorable impact on the systemic imbalance of the lipid profile, motor symptoms, and longevity in *Mecp2* (methyl CpG binding protein 2) mutant mice [72].

As many of these studies are small-scale or conducted in animal models, larger, well-controlled clinical trials are needed to establish the efficacy and safety of statin therapy in pediatric neurological disorders.

## 6. Statins and Oncologic Disorders

Recent studies suggest that statins may have anti-cancer properties, including inhibiting tumor cell proliferation, inducing cell death, and reducing inflammation [73]. As conventional treatment modalities, such as surgery, radiation therapy, and chemotherapy, have limitations in terms of efficacy and potential side effects, there is a need for alternative therapeutic strategies, and statins have emerged as a potential avenue for exploration. 

In children, medulloblastoma is the most prevalent malignant brain tumor, accounting for approximately 20% of all childhood brain tumors [74]. More treatment options are welcomed as recent studies revealed that medulloblastoma developed resistance to several commonly used chemotherapeutic agents [75]. Simvastatin, which acts as a mevalonate cascade inhibitor and passes the blood—brain barrier, induced apoptosis in cell lines derived from medulloblastoma brain tumors by a mechanism linked to the prenylation intermediates of the cholesterol metabolism pathway [76]. Fan et al. suggest that simvastatin is able to suppress the Hedgehog (Hh) pathway, whose activation may result in medulloblastoma, and, unlike other inhibitors of this pathway, such as sonedigib and vismodegib, effectively suppressed tumor growth in young mice, without causing any defects in bone development [77]. In humans, a phase 1 study conducted in children with relapsed/refractory solid and central nervous system tumors reported that plasma interleukin 6 (IL-6) concentrations consistently decreased over time, returning to normal values in all patients after three weeks of simvastatin, in association with topotecan and cyclophosphamide on days 1–5 [78]. Due to the favorable safety profile, affordability, and widespread accessibility of cholesterol inhibitors, directing efforts towards inhibiting cholesterol biosynthesis holds great promise as a treatment strategy for medulloblastoma and potentially other malignancies associated with hedgehog pathway activation [79].

The most common group of the pediatric central nervous system are gliomas, accounting for approximatively 45% of tumors [74]. Research suggests that glioma cells have the ability to convert cholesterol into corticosteroids such as progesterone, androstenedione, and androstenediol [80]. This conversion process has been associated with the potential acceleration of glioma progression [80].

Furthermore, in the era of an increasing prevalence of colon polyps and colorectal neoplasia [81], with an alarming nearly double incidence in young adults since 1990 [82], statins are beginning to be also viewed as chemo-preventive agents [83]. While preclinical and limited clinical evidence suggests potential benefits, further research is needed to establish the efficacy and safety of statins in the oncology field. The multifaceted mechanisms of action of statins make them an intriguing therapeutic option, and ongoing studies are waited to provide valuable insights into their potential role in improving outcomes for children with neoplastic conditions. Future research should focus on elucidating the optimal dosing, timing, and duration of statin therapy in children with various tumors. The identification of predictive biomarkers that can help identify patients who are most likely to benefit from statin treatment is crucial. Furthermore, exploring the potential combination of statins with other targeted therapies holds promise for enhancing treatment outcomes.

The potential chemo-preventive and therapeutic efficacy of different formulations of statins, including simvastatin, pitavastatin, and lovastatin, were examined in several preclinical studies [84,85,86]. For instance, Kubatka et al. showed that fluvastatin significantly decreased the frequency of mammary carcinoma by 63% in female rats [87]. The same study revealed that simvastatin significantly suppressed tumor frequency in a mammary carcinoma model by 80.5% and tumor incidence by 58.5% in comparison to the controls [87]. In a rat bladder carcinogenesis model, the tumor incidence was 65% with a mean tumor volume of 112.5  ±  6.4 mm^3^ in contrast to data from the group receiving atorvastatin, which was associated with a tumor incidence of 12.5% and mean tumor volume of 2.3  ±  0.2 mm^3^ [88]. In humans, the results of the available observational and randomized studies are controversial and insufficient to establish a consensus regarding the use of statins as chemo-preventive drugs in children [89].

## 7. Future Perspectives

The future perspectives of statin use in children are promising, considering their pleiotropic effects in various diseases. While the primary indication for statins in pediatric populations remains the management of dyslipidemia, their potential applications in autoimmune and inflammatory diseases, neurological disorders, and oncologic diseases warrant further investigation. 

Future research should focus on optimizing dosing, evaluating long-term safety, and conducting well-designed clinical trials to establish the efficacy and safety of statins in children with various conditions. In the context of pediatric cancers, the use of statins is still in its early stages, and more research is needed to fully understand their efficacy and safety in this specific population. Future studies may focus on investigating the potential benefits of statins as adjuvant therapy in combination with standard cancer treatments, such as chemotherapy or radiation therapy. 

The potential of statins to prevent cancers is also an area of ongoing research and investigation. While some studies suggest that statins may have a chemo-preventive effect and reduce the risk of certain cancers, the evidence is not yet conclusive. Further research, including large-scale clinical trials and long-term studies, is needed to better understand the role of statins in cancer prevention.

Also, it would be valuable to examine the mechanisms of action through which statins may exert their effects on neurologic conditions in children, as well as the optimal dosage and duration of treatment. As the prevalence of autism in children has shown an increasing trend over the past few decades [90], long-term follow-up studies are needed to assess the safety and efficacy of statins in this specific context.

Statin therapy has proven to be highly effective in reducing cardiovascular events and mortality in patients with dyslipidemia. However, there is considerable heterogeneity in the treatment response among patients, highlighting the need for personalized medicine approaches. The identification of specific biomarkers that can predict treatment response to statins is crucial for optimizing patient outcomes and resource allocation. It would be valuable to assess the significance of specific biomarkers regarding the suitable moment of treatment initiation, the early identification of potential complications, and the effectiveness in targeting specific diseases. For instance, endothelial function markers such as endothelial nitric oxide synthase (eNOS) and soluble intercellular adhesion molecule-1 (sICAM-1) can provide insights into the effects of statins on endothelial function and vascular health [91]. Moreover, genetic biomarkers, such as single nucleotide polymorphisms (SNPs) in genes involved in lipid metabolism, have shown promise in predicting statin response [92]. The role of specific genes as strong candidate genes including *CETP*, *HMGCR*, *SLCO1B1*, *ABCB1*, and *CYP3A4*/*5* should be further studied in relation with treatment outcomes [93]. By incorporating genetic, lipid, inflammatory, and metabolic biomarkers into clinical decision-making, clinicians can better tailor statin treatment to individual patients. Further research and validation of these biomarkers are warranted to establish their clinical utility and integration into routine practice. This personalized approach has the potential to improve treatment outcomes, reduce adverse effects, and optimize resource allocation. 

## 8. Conclusions 

The pleiotropic effects of statins sparked interest in exploring their potential use in various diseases. In addition to lipid-lowering effects, statins have been shown to have anti-inflammatory, antioxidant, and immunomodulatory properties. In children, HMG-CoA reductase inhibitors showed incontestable benefits with minimal side effects in various conditions associated with disturbances in lipid profile parameters. Besides the role of statins in hypercholesterolemia, innovative potential indications include neurological, oncologic, and cardiovascular diseases, as well as special situations such as transplanted children. While the use of statins in children for these conditions is still being studied, their pleiotropic effects offer promising possibilities for expanding their therapeutic applications beyond cholesterol management. However, it is important to note that further research is needed to fully understand the safety and efficacy of statins in these contexts before widespread use can be recommended.

## Figures and Tables

**Table 1 children-11-00117-t001:** Statins approved in children with hypercholesterolemia.

Statin	Age	Dose	Indication	Expected LDL-C % Reduction	Clinical Evidence of LDL-C % Reduction
Rosuvastatin	8–9 years	5–10 mg	HeFH	44–52%	38–50% in children aged 14.5 ± 1.8 years old treated with gradual dose-escalation up to 20 mg/day [11]
	10–17 years	5–20 mg		44–63%	
Pravastatin	8–13 years	20 mg	HeFH	30%	32% in 214 children aged 13 ± 2.9 years old treated with gradual dose-escalation up to 40 mg/day [12]
	14–18 years	40 mg		36%	
Atorvastatin	10–17 years	10–20 mg	HeFH	37–60%	39.9–43.8% in children aged 6–15 years old treated with gradual dose-escalation up to 80 mg/day [13]
	≥10 years	10–80 mg	HoFH	37–60%	
Fluvastatin	10–16 years	20–80 mg	HeFH	20–36%	34% in children aged 10–16 years old treated with 80 mg/day [14]
Simvastatin	10–17 years	10–40 mg	HeFH	30–40%	41% in children aged 14.4 ± 2.1 years old treated with gradual dose-escalation up to 40 mg/day [15]
Lovastatin	10–17 years	10–40 mg	HeFH	20–30%	17–37% in girls aged 14.4 ± 2.1 years old treated with gradual dose-escalation up to 40 mg/day [16]

HeFH—Heterozygous familial hypercholesterolemia, HoFH—homozygous familial hypercholesterolemia; LDL—low-density lipoprotein cholesterol.

**Table 2 children-11-00117-t002:** Main follow-up studies on children with familial hypercholesterolemia treated with statins.

Authors, Publication Year	Follow-Up Duration	Population Size and Age *	Statin	Outcomes	Side Effects
Luirink et al., 2019 [12]	20 years	214 children (mean age: 13 ± 2.9 years old)	Pravastatin	LDL ↓ 32% Cardiovascular events ↓ (1% in intervention group vs. 26% in affected parents) Death from cardiovascular causes ↓ (0% vs. 7%, respectively)	No reported cases of rhabdomyolysis or any other serious adverse events. No apparent adverse effect on general development or growth.
Mamann et al., 2019 [26]	4 years **	131 (5.2–17.7 years old, mean age: 10 years old)	Pravastatin (101 cases), Rosuvastatin (22 cases), Atorvastatin (8 cases)	TC ↓ 24.4% LDL ↓ 32% HDL ↑ 6% TG ↓ 1.6%	No apparent adverse effect on general development or growth. In total, 24 (18.4%) patients had minor adverse events (5 cases with ↑ CPK levels, 14 cases of myalgia, 3 cases of abdominal pain, 1 case of dysuria, and 1 case of diffuse pain).
Langslet et al., 2016 [13]	3 years	272 (6–15 years old)	Atorvastatin	LDL ↓ 43.8% TS 1 and 39.9% for TS ≥2	Six (2.2%) subjects discontinued because of adverse events. No apparent adverse effect on general development or growth.
Carreau et al., 2011 [27]	2 years 2 months	185 (mean age: 11 years old)	Pravastatin	TC ↓ 16.9–19.2% LDL ↓ 21–24%	Mostly minor side effects reported in 13% of children, 4 cases with muscular symptoms that could be treatment-related.
Avis et al., 2010 [11]	1 year 1 month	176 children (mean age: 14.5 ± 1.8 years old) with HeFH	Rosuvastatin	TC ↓ 27.1% LDL ↓ 33.9% TG ↓ 5.3% ApoB ↓ 24.2%	No hepatic, skeletal muscle, or renal side effects leading to permanent treatment discontinuation. No apparent adverse effect on general development or growth.

* Age at baseline, ** median duration, LDL—Low density lipoprotein, TC—total cholesterol level, TS—Tanner stage, CPK—creatinphosphokinase, CVD—cardiovascular disease. Arrow down signifies decreased, arrow up signifies increased.

**Table 3 children-11-00117-t003:** Secondary causes of hypercholesterolemia in children (adapted from [8]).

Class of Diseases	Specific Disease
Endocrine/Metabolic	Diabetes, obesity, hypothyroidism, glycogen storage diseases, Niemann-Pick disease, Gaucher disease, polycystic ovarian syndrome, lipodystrophy, pregnancy
Cardiac	Kawasaki disease, orthotopic heart transplant
Rheumatological	Juvenile inflammatory arthritis, systemic lupus erythematous
Gastrointestinal	Alagille syndrome, biliary cirrhosis, hepatitis, obstructive liver disease
Renal	Nephrotic syndrome, chronic renal disease, renal transplant,
Infectious	Acute viral or bacterial infection, human immunodeficiency virus, hepatitis
Medication	Glucocorticoids, isotretinoin, oral contraceptives, antipsychotics, alcohol, betablockers
Other	Anorexia nervosa, solid organ transplantation, childhood cancer survivors

## Data Availability

Data are contained within the manuscript.

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
