# Peer review of "Statins—Beyond Their Use in Hypercholesterolemia: Focus on the Pediatric Population"

_children, 2024, doi:10.3390/children11010117_

Round 1
Reviewer 1 Report
Comments and Suggestions for Authors
It is a good review about statins in pediatric age. I recommend revision of few issues.
- Inflammation is ameliorated by statins and this issue must be elaborated better in the review. Statins have important role in vasculitic disorders (https://doi.org/10.3109/08916934.2015.1027818),
- Adverse events caused by statins can be emphasized better. They may cause liver test abnormalities, myositis and even rhabdomylysis (https://doi.org/10.1016/j.cjca.2020.03.041).
- Safety issues and drug interactions of statins in children can be commented on, too. I recommend a recent publication about this topic (https://doi.org/10.3390/ijms24021366).
Author Response
Dear Reviewer,
Thank you for your valuable time and input. We believe your comments have helped improved our manuscript.
Hereby we provide a point by point response.
It is a good review about statins in pediatric age. I recommend revision of few issues.
- Inflammation is ameliorated by statins and this issue must be elaborated better in the review. Statins have important role in vasculitic disorders (https://doi.org/10.3109/08916934.2015.1027818)
Thank you for your recommendation. Indeed, the statins have been shown to have potential anti-inflammatory effects in vasculitic disorders. We added new details about this issue in lines 208-222.
- Adverse events caused by statins can be emphasized better. They may cause liver test abnormalities, myositis and even rhabdomylysis (https://doi.org/10.1016/j.cjca.2020.03.041).
We agree. We revised the content related to adverse events particularly in children. Please see the lines 105-119.
- Safety issues and drug interactions of statins in children can be commented on, too. I recommend a recent publication about this topic (https://doi.org/10.3390/ijms24021366).
Thank you for your suggestion regarding the safety issues and drug interactions of statins in children. Please see the lines 131-153.
Reviewer 2 Report
Comments and Suggestions for Authors
Dear author,
I have studied with great interest the manuscript “Statins- beyond their use in hypercholesterolemia: focus on paediatric population”.
The main question addressed by the research was to cover the information about the usage of statins in children.
The topic is original. The manuscript is clearly exposed and well written. The figure and tables correspond to the description in the text, are well designed and reflect important information. The references are appropriate.
However, I have some comments to improve the quality of the paper.
1. The future perspectives should be provided.
I express my gratitude to the authors for their great work done.
Author Response
Dear Reviewer,
Thank you for your valuable time and input. We believe your comments have helped improved our manuscript.
Hereby we provide a point by point response.
Dear author,
I have studied with great interest the manuscript “Statins- beyond their use in hypercholesterolemia: focus on paediatric population”.
The main question addressed by the research was to cover the information about the usage of statins in children.
The topic is original. The manuscript is clearly exposed and well written. The figure and tables correspond to the description in the text, are well designed and reflect important information. The references are appropriate.
However, I have some comments to improve the quality of the paper.
The future perspectives should be provided.
I express my gratitude to the authors for their great work done.
Thank you very much for your valuable comments. We have now provided the future perspectives on this topic. Please see Section 6, lines 368-410.
Reviewer 3 Report
Comments and Suggestions for Authors
This manuscript presents an important topic of pleiotropic effects of statins in paediatric population. The paper is well written, the language used throughout the text seems correct and the references are in general up-to-date.
However, authors are encouraged to add 2-3 lines-sentences on the conversion of cholesterol into corticosteroids process associated with the potential acceleration of glioma progression, and that statins are beginning to be aso viewed as chemopreventive agents, elucidating the dark sides through the existing bibliography. Authors should aso add 2-3 sentences on the crucial role of several specific biomarkers identifying patients who are most likely to benefit from statin treatment in order to better and most precisely describe this interesting clinical model.
Author Response
Dear Reviewer,
Thank you for your valuable time and input. We believe your comments have helped improved our manuscript.
Hereby we provide a point by point response.
This manuscript presents an important topic of pleiotropic effects of statins in paediatric population. The paper is well written, the language used throughout the text seems correct and the references are in general up-to-date.
However, authors are encouraged to add 2-3 lines-sentences on the conversion of cholesterol into corticosteroids process associated with the potential acceleration of glioma progression, and that statins are beginning to be aso viewed as chemopreventive agents, elucidating the dark sides through the existing bibliography.
Thank you for pointing this out. We have added this information in lines 337-341, as well as 355-366.
Authors should aso add 2-3 sentences on the crucial role of several specific biomarkers identifying patients who are most likely to benefit from statin treatment in order to better and most precisely describe this interesting clinical model.
Thank you for this suggestion. We made the modifications accordingly. Please see the lines 391-410.